# Retrospective and statistical analysis of hand and forearm injuries in the Silesian pediatric population – study of post-traumatic X-rays in 2022

**Magdalena Machnikowska-Sokołowska**[1]☯, **Marcin Ciekalski** [2]☯*, **Iga Szymańska**[2]‡, **Jakub Mordarski**[2]‡, **Katarzyna Gruszczyńska**[3]

**1** Department of Diagnostic Imaging, Radiology and Nuclear Medicine, Medical University of Silesia, Katowice, Poland, **2** Students Scientific Society, Department of Radiology and Nuclear Medicine, Medical University of Silesia, Katowice, Poland, **3** Department of Radiology and Nuclear Medicine, Medical University of Silesia, Katowice, Poland

☯ These authors contributed equally to this work.
‡ These authors also contributed equally to this work.
* s76402@365.sum.edu.pl

## Abstract

### Background and objectives

Upper limb injuries are a common occurrence in the pediatric demographic, particularly wrist injuries, which constitute a noteworthy subset. Fractures involving the forearm, wrist, and phalanges are recurrent reasons prompting visits to emergency departments. This investigation endeavors to offer a comprehensive insight into hand and forearm fractures within the pediatric cohort, delineating data across various age groups, gender distributions, and fracture location.

### Materials and methods

This study involved data from the reports of 834 post-traumatic wrist X-ray examinations conducted in 2022 at the Department of Diagnostic Imaging of the Upper Silesian Child Health Centre. The analysis included bone injuries, the age and gender of the child, the nature of the lesions, and anatomical location of the injury. The study group consisted of 327 girls and 511 boys, who were on average 11.4 ± 3.6 and 12.0 ± 3.8 years old, respectively.

### Results

There were 273 single fractures and 66 multiple fractures in the study group. The highest number of fractures were recorded in the metacarpal bones (29,2%), among which the fifth meta-carpal bone was the most frequently injured (14,5%). The second most common fracture was in the forearm (28,9%). There were 55 phalangeal fractures in the study group (16,2%).

**Data availability statement:** Data cannot be shared publicly because of potentially identifying or sensitive patient information. Data set can be shared upon request from the Medical University of Silesia Institutional Data Access for researchers who meet the criteria for access to confidential data. For data acquisition, please contact the Administration Office of our Radiology Department at zdo@gczd.katowice.pl.

**Funding:** The author(s) received no specific funding for this work.

**Competing interests:** The authors have declared that no competing interests exist.

## Conclusions

The risk of fractures is statistically highest among boys, peaking at 12 years of age. The metacarpal bones and the radius are most susceptible to fractures. The likelihood of fractures in the right and left hand is similar in females, whereas males are more likely to sustain injuries to the right hand.

## Introduction

Hand and forearm injuries constitute a significant subset of orthopedic traumas in pediatric populations, with the potential to impact the quality of life of young individuals. These injuries can arise from various causes including accidents, sports-related incidents, or other unfortunate events. The highest risk of fractures is among children between 10 and 14 years of age [1]. The overall incidence of pediatric distal forearm fractures is 738.1/100,000 persons/year and constitutes 74% of all pediatric fractures in the upper limb [2,3].

Despite ongoing endeavors to enhance the safety of children and adolescents, there is a notable and statistically significant rise in the occurrence of hand and forearm fractures within this demographic [4]. This issue presents both clinical and socioeconomic challenges due to parents' absence from work, treatment costs, and hospital stays. Scaphoid fractures, the most common carpal bone fracture, typically occur after a fall onto an outstretched hand, with patients often presenting with wrist pain following the injury. However, diagnosing scaphoid bone fractures on initial radiographs can be difficult. If suspicion arises regarding a scaphoid fracture and initial radiographs yield inconclusive results, it is advisable to conduct a dedicated x-ray specifically targeting the scaphoid bone. Persistent clinical symptoms warrant a follow-up x-ray examination within 7–14 days. If these x-rays remain negative for a scaphoid fracture, Magnetic Resonance Imaging (MRI) should be considered for further diagnostic evaluation [5]. On the other hand, computed tomography (CT) has been found to be more dependable and accurate in evaluating the characteristics of a scaphoid fracture and the healing process [6]. Bearing in mind the ALARA ("as low as reasonably achievable") principle, it is recommended to avoid the CT examination to reduce harmful radiation exposition. It is possible thanks to the increasingly available sequences dedicated to the evaluation of bone structures in MR examinations.

Currently, the availability of consistent, detailed, and up-to-date epidemiological data on hand and forearm injuries in the pediatric population in Poland is limited. The primary objective of this study was to conduct a comprehensive retrospective and statistical analysis of hand and distal forearm injuries in the Silesian pediatric population. Our research will serve as a valuable source of information for the medical community and parents to develop more effective diagnosis, preventive strategies and ensure better care for children and adolescents at risk of hand and distal forearm injuries in the Silesian region and beyond.

In the pediatric population of Silesia, Poland, fractures are a significant concern, particularly among adolescents. This study identifies key trends in fracture prevalence and patterns. The main contributions of this work are:

- Fractures are most prevalent among adolescent males, with a peak incidence at 12 years of age.

- The fifth metacarpal bone is the most frequent site of fracture.

- While females exhibit an equal likelihood of fracturing either hand, males demonstrate a higher propensity for injuring their right hand.

## Materials and methods

### Study design, area, and period

The study involved data from the reports of 834 post-traumatic hand and distal forearm X-ray examinations performed from January to December 2022 at the Department of Diagnostic Imaging of the John Paul II Upper Silesian Child Health Centre in Katowice, Poland. The Ethics Commission of the Medical University of Silesia in Katowice exempted the need for ethical approval for this study. Data from the hospital system was shared with researchers on February 15, 2023. Patient records were acquired as an anonymized report from the hospital's internal database, making it impossible to identify individual patients based on the gathered data. Most of the patients were Polish, but there were 37 Ukrainians, 3 Russians, 2 Belarusians, 1 Korean, Turkish, American, Serbian, Chinese, Iranian.

The analysis comprised hand X-rays captured in posterior-anterior (PA) and lateral projections, supplemented in certain instances by dedicated x-rays focusing on the scaphoid bone. X-ray studies were identified and retrieved from the computer system based on the ICD (International Classification of Diseases) code and the type of examination. In our center, hand x-rays cover the area of the distal forearm, wrist, metacarpal bones, and phalanges.

The study recorded the seasonal distribution, anatomical localization of the injury, presence of fracture, as well as the patient's age and gender. An injury is defined as any damage or harm to the body caused by external physical force or trauma, whereas a fracture specifically refers to a disruption in the continuity of a bone. The study does not contain data regarding elbow dislocation. The primary outcome measures were the count and ratios of injuries and fractures. Gender, age and the month of injury occurrence were employed as independent variables for classification. Patients were categorized into five age groups: 0–4 years, 5–8 years, 9–12 years, 12–16 years, and 17–18 years. The age categorization is based on the study Naranje SM et al. Epidemiology of Pediatric Fractures Presenting to Emergency Departments in the United States and modified for maximum age of 18.

According to the Act of December 5, 1996, on the on the professions of doctor and dental practitioner (i.e., Journal of Laws of 2023, item 1516 as amended), retrospective analysis of medical data is not a medical experiment and does not require the evaluation of the Bioethics Committee of the Medical University of Silesia. Despite the aforementioned legal act, the Bioethics Committee of our university issued an opinion (BNW/NWN/0052/KB/294/23/24) stating that our study does not require official approval from the Bioethics Committee.

### Statistical analysis

The collected data underwent statistical analysis using Statistica 12.0 software (StatSoft, Kraków, Poland). Normality of distribution was assessed using the Shapiro-Wilk test, revealing that none of the variables, except age, exhibited a normal distribution. The U Mann–Whitney test was employed to assess differences in quantitative variables between two groups, while the Chi-squared test was used to compare differences in categorical variables between the two groups. Kruskal-Wallis test was utilized for comparing across multiple groups. A significance level of $p < 0.05$ was applied to all tests.

## Results

The study group consisted of 327 girls and 511 boys who were 11.4 ± 3.6 and 12.0 ± 3.8 years old, respectively (min. 0, max. 18 years). Table 1. displays the number of single fractures according to their locations for the entire study group, categorized by gender.

Table 1. Number of single fractures total and by gender.

| Localization | All (*n* = 273) | Girls (*n* = 71) | Boys (*n* = 202) |
|---|---|---|---|
| Forearm | 98 | 31 | 67 |
| Radius | 87 | 29 | 58 |
| Radial styloid process | 1 | 1 | 0 |
| Distal radial metaphysis | 84 | 28 | 56 |
| Shaft of the radius | 2 | 0 | 2 |
| Ulna | 11 | 2 | 9 |
| Ulnar styloid process | 4 | 1 | 3 |
| Distal ulnar epiphysis | 5 | 1 | 4 |
| Shaft of the ulna | 2 | 0 | 2 |
| Carpus | 21 | 9 | 12 |
| Scaphoid bone | 21 | 9 | 12 |
| Metacarpus | 99 | 16 | 83 |
| I | 18 | 4 | 14 |
| II | 14 | 1 | 13 |
| III | 9 | 1 | 8 |
| IV | 9 | 1 | 8 |
| V | 49 | 9 | 40 |
| Phalanges | 55 | 12 | 43 |
| I | 5 | 0 | 5 |
| II | 9 | 1 | 8 |
| III | 8 | 1 | 7 |
| IV | 8 | 4 | 4 |
| V | 25 | 6 | 19 |

The primary site of injury was the distal metaphysis of the radius and ulna (S1 and S2 Figs). Among the metacarpal bones, the fifth bone displayed the highest frequency of fractures (S3 Fig). Regarding phalanges, the fifth phalanx exhibited the highest frequency of injury (S4 Fig). A statistically significant difference was observed between males and females regarding all injuries ($p < 0.001$), as well as single fractures ($p < 0.001$) and fractures involving multiple sites ($p < 0.001$) (Fig 1). Multiple fractures were most commonly observed in boys (81.8%) with a mean age of 10.8 ± 4.6 years. The distribution of fractures by age groups and gender is detailed in Table 2.

There was no significant difference in the injury-to-fracture ratio based on age group ($p = 0.678$), affected side ($p = 0.936$), or the specific month when the injury took place ($p = 0.925$). However, within the fracture group, there was a significant age difference between males and females ($p < 0.001$), indicating that older males were more prone to experiencing fractures (Fig 2).

A significant difference was noted with respect to age group and fracture location ($p < 0.001$) (Fig 3).

The distribution of fractures, categorized by the month of occurrence and separated by gender, is illustrated in Fig 4. There were no significant differences in the occurrence month for either hand injuries ($p = 0.926$) or fractures ($p = 0.926$). Additionally, when considering age within the fracture group, no noteworthy differences were observed in terms of the month of occurrence ($p = 0.184$), as shown in Fig 5.

A significant difference exists between males and females concerning the site (right or left) that was fractured. In males, there was a statistically significant tendency to injure the right hand ($p = 0.01$), while no statistically significant difference was observed in females (Fig 6).

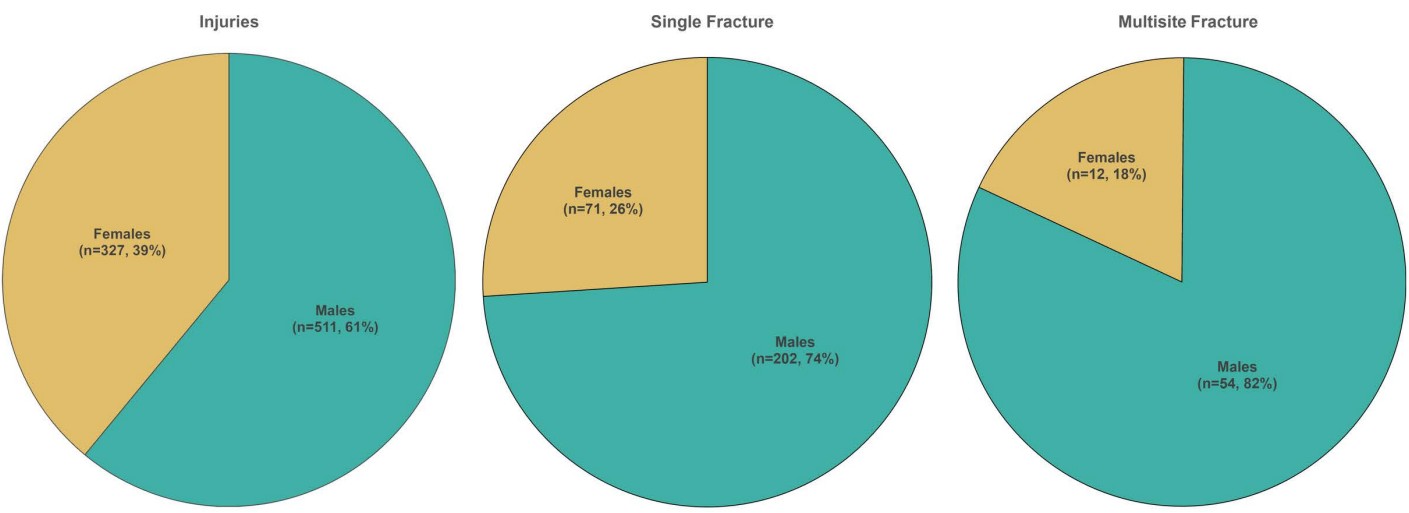

**Fig 1. The ratio of injuries to single and multisite fractures distributed by gender.**

**Table 2. Distribution of injuries and fractures by age group and gender.**

| Age (Years) | Injuries, N (F/M) | % | Fractures, N (F/M) | % |
|---|---|---|---|---|
| 0–4 | 55 (19/36) | 6.59 | 16 (1/15) | 4.72 |
| 5–8 | 77 (32/45) | 9.23 | 39 (12/27) | 11.50 |
| 9–12 | 309 (138/171) | 37.05 | 123 (45/78) | 36.28 |
| 13–16 | 333 (117/216) | 39.93 | 138 (24/114) | 40.70 |
| 17–18 | 60 (16/44) | 7.19 | 23 (1/22) | 6.78 |
| Total | 834 (322/512) | 100 | 339 (83/256) | 100 |

There were only 22 scaphoid bone fractures, accounting for 2.64% of all injuries and 6.49% of all fractures (Table 3, S5 and S6 Figs).

## Discussion

The prevalence of hand fractures varies across diverse geographic regions, creating the need to determine it for children's populations [1,3,7,8]. These documented distinctions may stem from disparities in ethnicity, culture, environmental influences leading to distinct activities, and variations in research methodologies.

The analysis of x-rays in this study revealed fractures in 33% of cases, a rate twofold higher than reported in other available studies (15–16%) [9–11]. The high occurrence of fractures in our study might reflect the preeminent quality of the preliminary assessment conducted by the pediatric surgeons in the Emergency Department, ensuring accurate determination of the indications for diagnostic imaging.

According to the available literature the most common fracture locations in pediatric patients are distal ulna, radius and distal humerus [12,13]. The standard diagnostic approach for forearm bone fractures involves acquiring x-ray images of the forearm, encompassing both the elbow and the radiocarpal joint, using anteroposterior (AP) and lateral orthogonal forearm radiographs (Fig 7) [14]. We analyzed all fractures within the scope of the hand radiograph, including injuries to the wrist, metacarpals, and fingers, as well as the distal parts of the

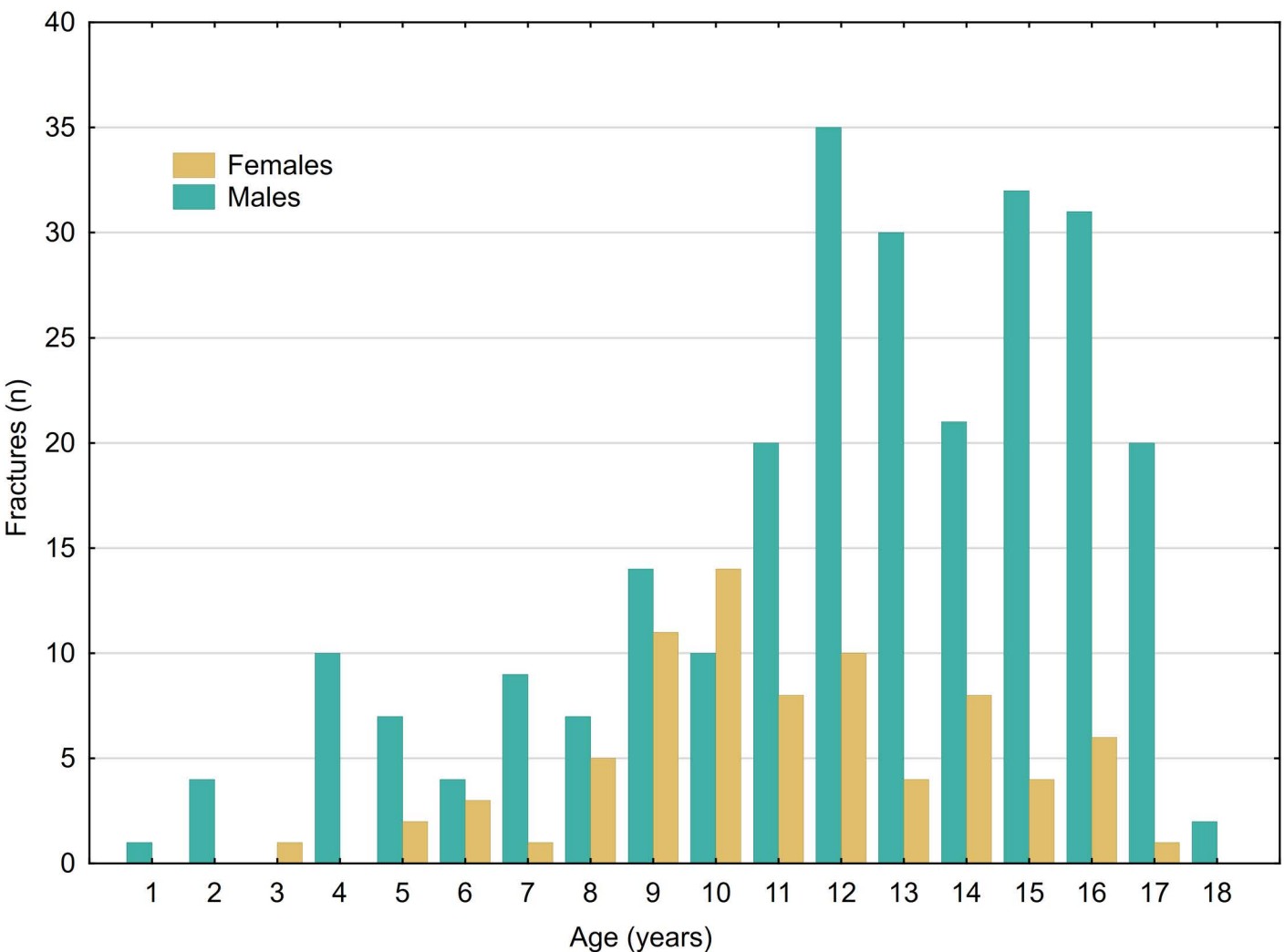

**Fig 2. Number of fractures distributed by age and gender.**

forearm bones. The methodology for wrist x-ray examination (S7 Fig) differs from that of the forearm (Fig 7). However, due to the prevalence of injuries in the distal part of the forearm, their analysis was integral and could not be omitted. In our study, the most common location of fracture were metacarpal bones (among which the fifth metacarpal was the most fractured). This result aligns with findings from other studies [15,16]. Nevertheless, certain studies have identified the fifth finger and proximal phalanx as the predominant sites of fracture [7,17]. Most studies concur that fractures are most frequent in the fifth finger and the metacarpophalangeal joint [7,9,15–18].

Although most of the population is right-handed, injuries affected both hands equally in females, whereas males exhibited a higher prevalence of right-hand injuries, consistent with findings from other studies [7,15,17,18].

In our investigation, most hand injuries were documented in males, constituting 61%, with the highest incidence observed within the age group of 9-12 and 13-16 years. These findings align with similar patterns reported in other research studies [7,19].

In our cohort, fractures were most observed in the end of spring and the beginning of autumn. This pattern could be associated with favorable weather conditions during these

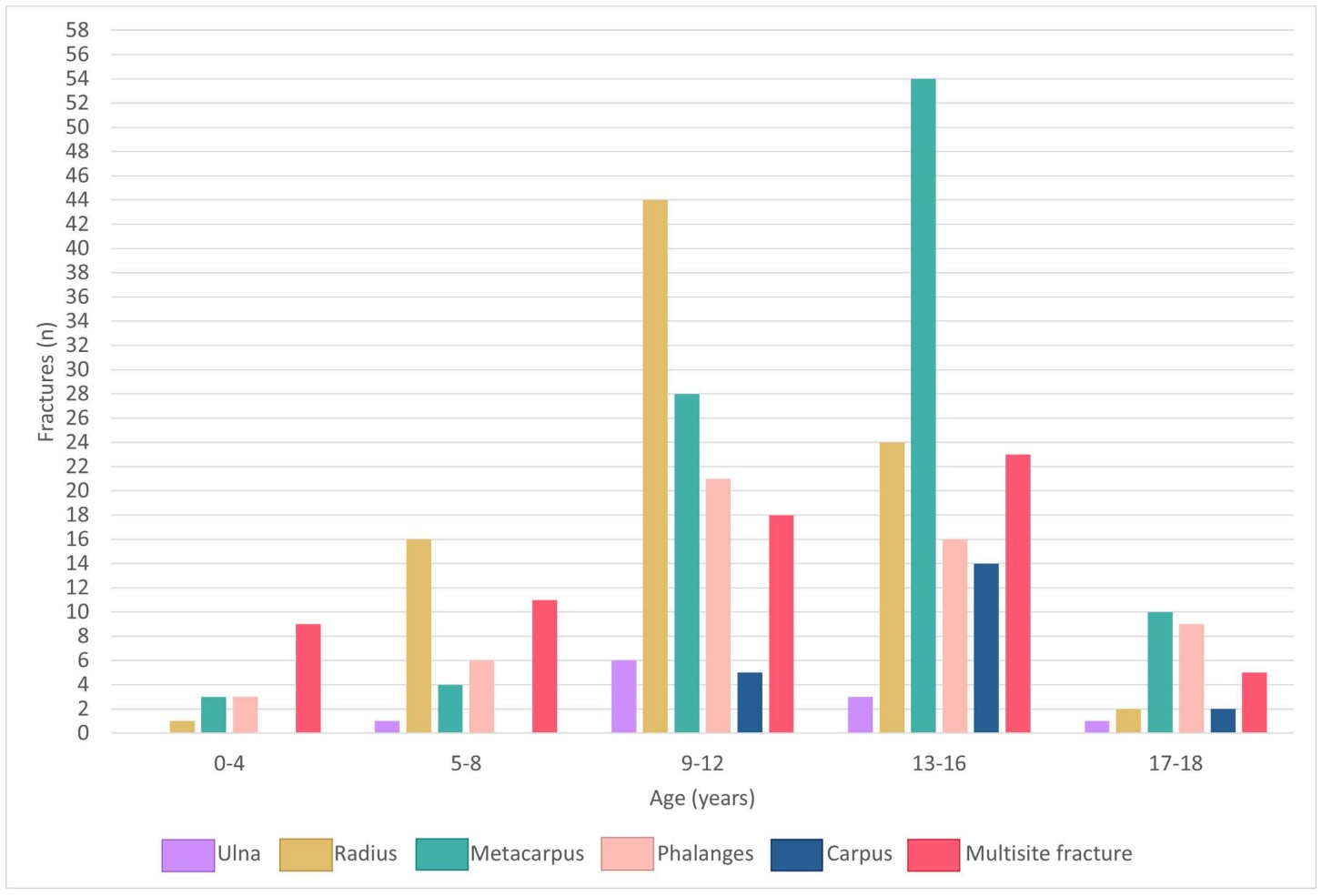

**Fig 3. Localization of fractures distributed by age group.**

seasons, encouraging increased engagement in sports activities both at school and outdoors. The Silesian voivodeship experiences a lower influx of tourists during the summer, as a considerable portion of the local population tends to spend their summer vacation outside our region. Notably, the school summer break in our area occurs in July and August, and the decline in fracture incidences during these months may be attributed to residents spending their summers in other parts of the country. The scientific literature regarding seasonal disparities in the frequency of hand injuries is limited. The consensus in most studies is that there are no significant differences in the occurrence of hand injuries across various months [20].

Based on our literature review, this study is the first to simultaneously address injuries of both the distal forearm and wrist. In previous studies that focused exclusively on the forearm, there was a potential for hand injuries to be overlooked. Conversely, in studies that considered only the hand, injuries to the forearm might have been missed.

The study's retrospective design and the potential for incomplete data, such as missing information on the cause of injury, absent records, or incomplete ICD codes, represent significant limitations. Furthermore, all data were obtained solely from a single pediatric facility. Despite these constraints, our detailed analysis of a large cohort of pediatric patients with hand fractures treated over a one-year period offers valuable insights.

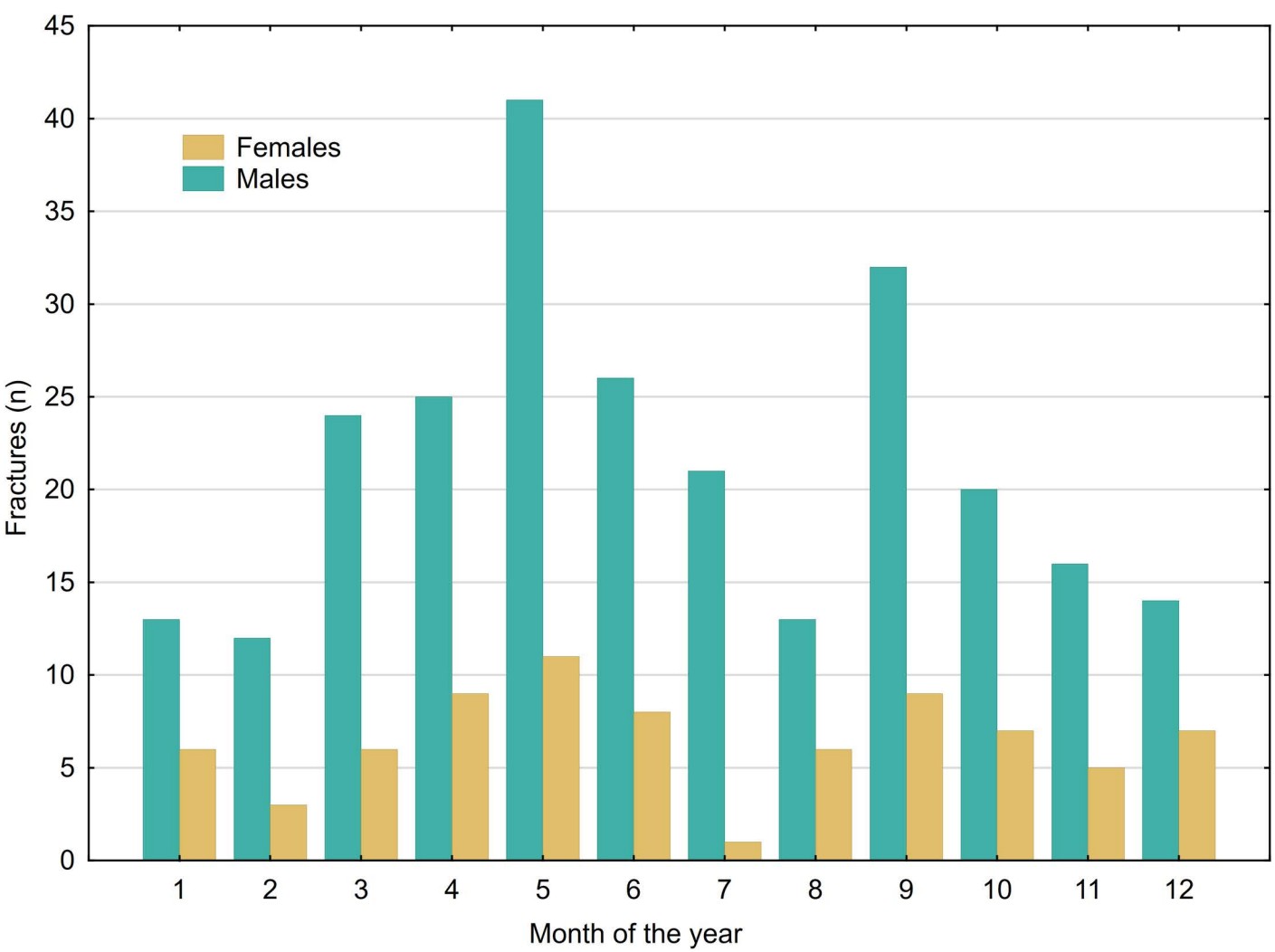

**Fig 4. Monthly distribution of fractures allocated for gender.**

## Conclusions

This study highlights significant patterns in pediatric hand fractures within the Silesian region of Poland. The findings indicate that fractures are most prevalent among adolescent males, peaking at 12 years of age. The fifth metacarpal bone emerges as the most frequent site of fracture. Additionally, a gender-based disparity is observed: females exhibit an equal likelihood of fracturing either hand, whereas males demonstrate a higher propensity for injuring their right hand.

These results provide valuable insights for targeted prevention strategies and resource allocation in pediatric care.

Recommendations for treatment and prevention of hand and forearm injuries:

1. Diagnosis

- On patient presentation, history should cover specifics like the mechanism and time of injury and the extent and duration of crushing force.

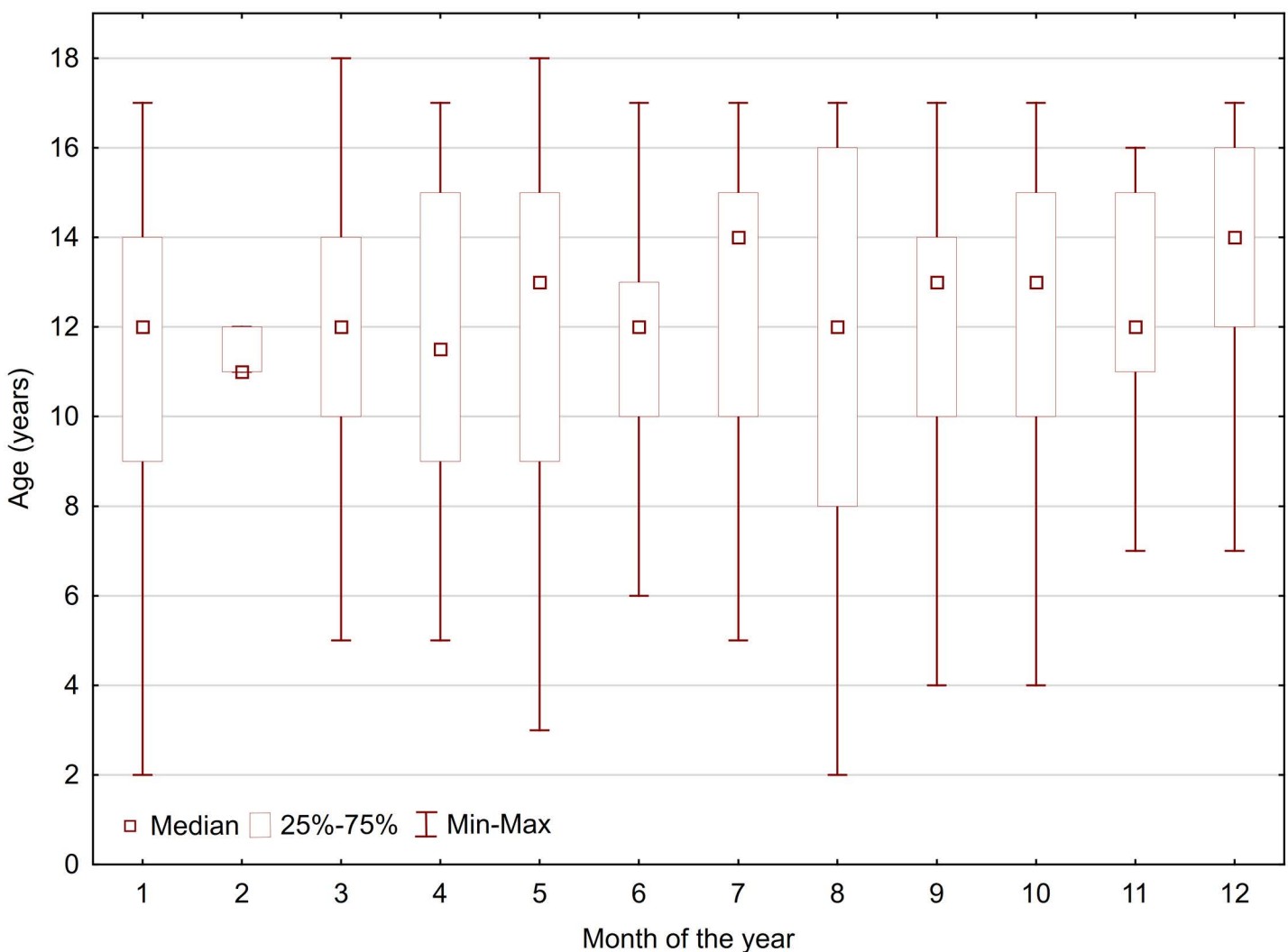

**Fig 5. Monthly distribution of fractures allocated for age.**

- It's crucial to assess whether there is a break in the overlying skin or a penetrating injury – open wounds should be inspected for the degree of contamination and any potential tendon, ligament or neurovascular injury [21].

- Hand examination is performed by comparing the injured hand to the uninjured hand, based on Apley's principles of "look, feel, move" [5,21,22].

- X-rays are the first-line imaging technique for fractures [21,23,24]. MRI and CT scans are recommended in cases of normal radiographs with clinical picture suspecting fracture [21,23].

2. Management

- Pain should be assessed at the first patient presentation using visual analogue scale (VAS) in younger children or numeric rating scale in older children. The assessment should be done repeatedly [21,25,26].

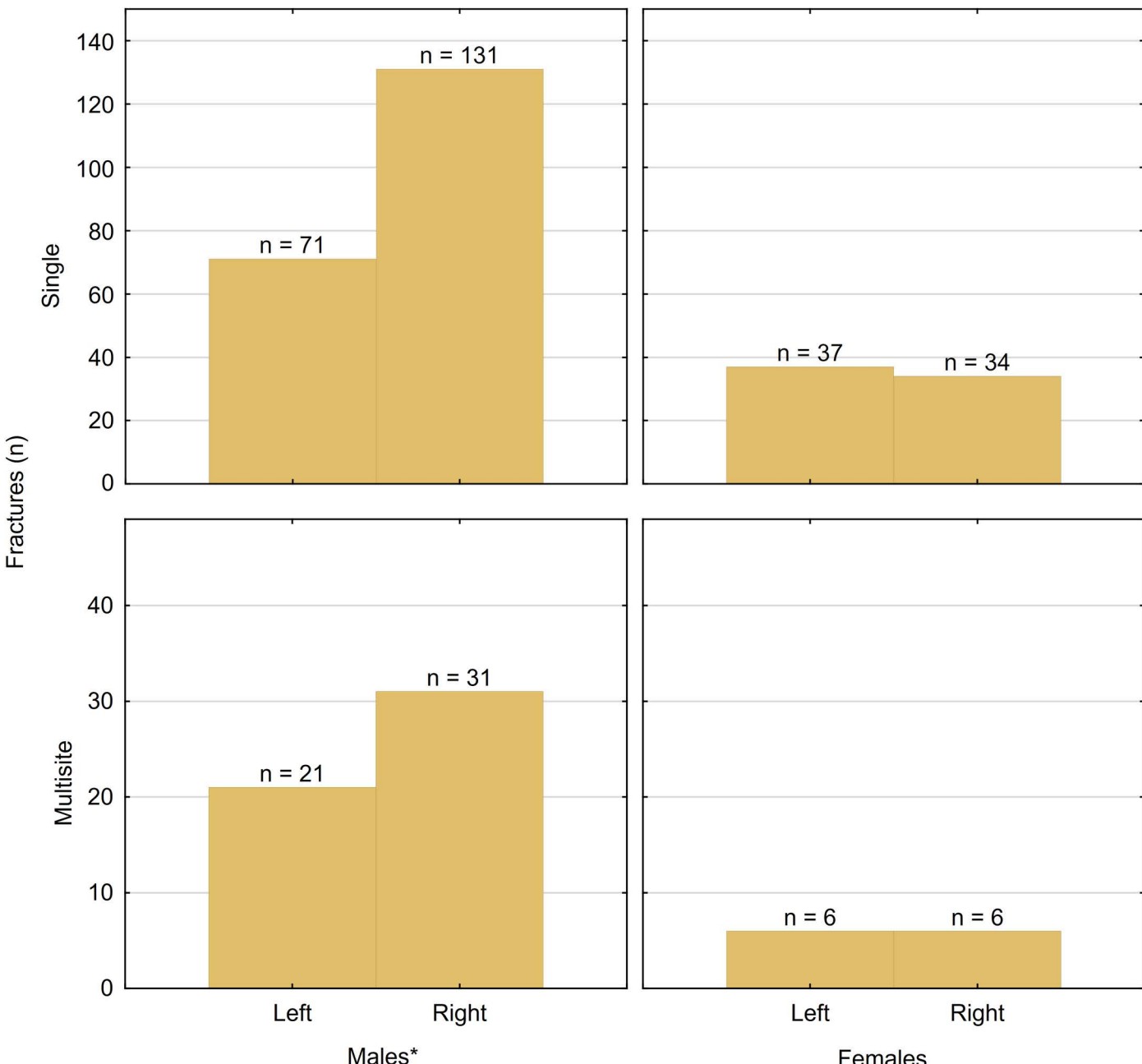

**Fig 6. Fractures distribution based on multitude, gender, and the site of the fracture.** * Two males sustained fractures in both of their hands.

- Pain management involves non-opioid oral medications such as ibuprofen or paracetamol or both for mild to moderate pain, and intravenous or intranasal opioids for moderate to severe pain [21,27,28].

- The majority of hand fractures can be managed non-surgically through closed reduction, splinting and early mobilization [21,29–31].

- Splinting should include the joint above and below the fracture and shouldn't exceed three to four weeks to avoid joint stiffness [21,32].

**Table 3. Distribution of scaphoid bone fractures by age group and gender.**

| Age group (years) | Cases, N (F/M) | % |
|---|---|---|
| 0-4 | 0 (0/0) | 0.00 |
| 5-8 | 0 (0/0) | 0.00 |
| 9-12 | 5 (1/4) | 22.73 |
| 13-16 | 10 (4/6) | 45.45 |
| 17-18 | 7 (5/2) | 31.82 |
| Total | 22 (10/12) | 100 |

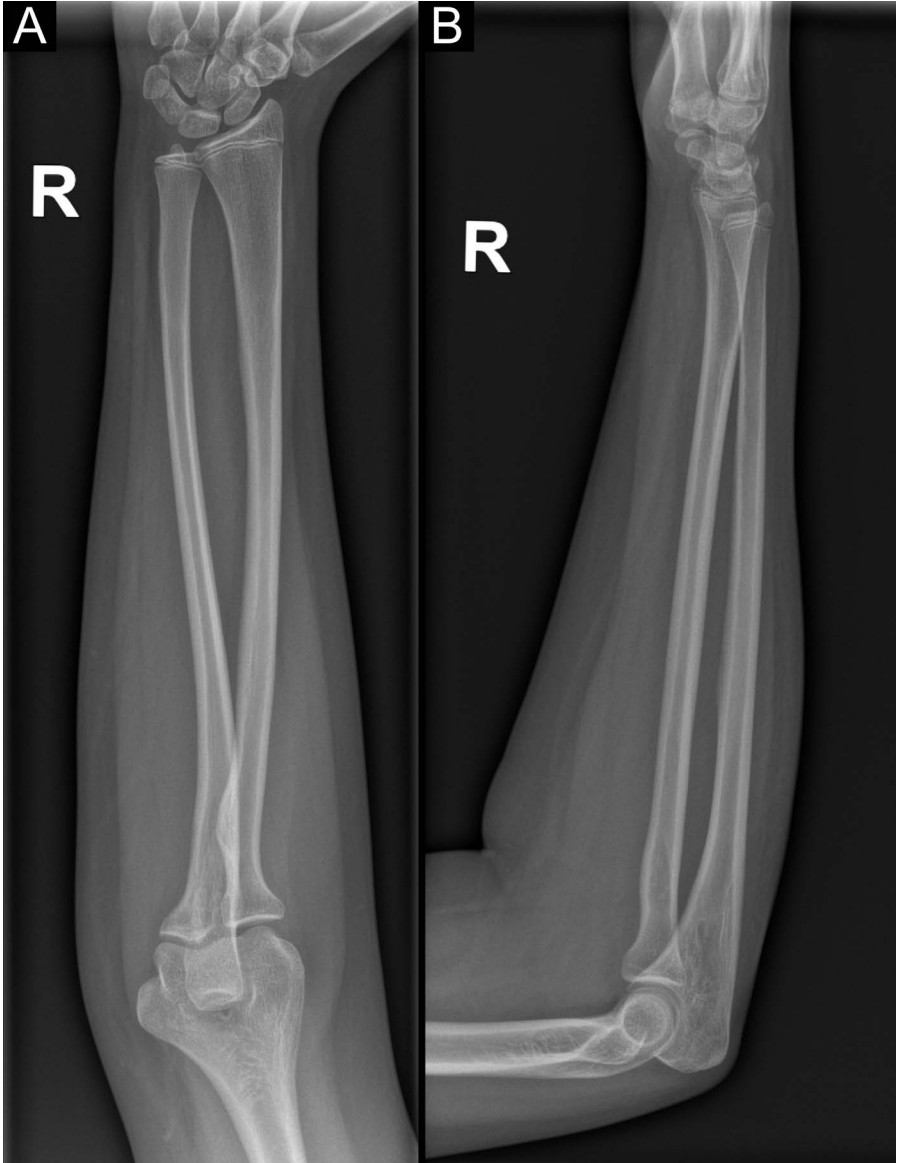

**Fig 7. The methodology of performing forearm radiography in PA (A) and lateral (B) projections, encompassing two adjacent joints (wrist and elbow).**

- The hand with splint should be elevated above the level of the heart to reduce swelling [21,32].

- The indications for surgical management of hand fractures: open or unstable fractures, significant displacement, digital rotation or metacarpal shortening, fractures involving multiple digits or injury of the soft tissue requiring reconstruction [21,32–36].

- Conservative treatment for forearm fractures is only indicated in a few cases, such as unicortical fractures and fractures that are either undisplaced or minimally displaced [24].

- Forearm fractures are typically treated with surgical fixation, which may involve either closed or open reduction and internal fixation [24].

3. Prevention:

- The strategy of "preserving bone mass, slowing down and reducing the likelihood of falling" is likely to reduce risk distal forearm fractures [37].

## Supporting information

**S1 Fig. PA (A) and lateral (B) forearm X-ray.** Torus-type fracture characteristic for children in the left distal radial metaphysis.
(TIF)

**S2 Fig. PA (A) and lateral (B) wrist X-rays.** Angulation of the cortical layer in the distal metaphysis of the left radius.
(TIF)

**S3 Fig. Oblique (A) and PA (B) projections of hand x-ray.** Multidirectional fracture of the shaft of the fifth metacarpal bone of the left hand with displacement of the fragment to the dorsal site.
(TIF)

**S4 Fig. PA (A) and oblique (B) projections of hand x-ray.** The injury under the angulation of the cortical layer at the base of the proximal phalanx of the fifth finger of the left hand. Visible ossification nuclei and growth cartilage.
(TIF)

**S5 Fig. Wrist x-ray, involving lateral (A), posterior-anterior (PA) (B), and a dedicated projection for the scaphoid bone (C).** The injury can be seen only in the form of unevenness of the cortical layer in the waist of the bone.
(TIF)

**S6 Fig. X-ray images of the wrist, captured in both the PA and lateral projections (B), with a dedicated view focused on the scaphoid bone (C).**
(TIF)

**S7 Fig. X-ray of the wrist in AP (A) and lateral (B) projections.**
(TIF)

## Author contributions

**Conceptualization:** Magdalena Machnikowska-Sokołowska, Marcin Ciekalski.

**Data curation:** Marcin Ciekalski.

**Formal analysis:** Marcin Ciekalski.

**Investigation:** Marcin Ciekalski, Iga Szymańska, Jakub Mordarski.

**Methodology:** Magdalena Machnikowska-Sokołowska, Marcin Ciekalski, Katarzyna Gruszczyńska.

**Project administration:** Magdalena Machnikowska-Sokołowska.

**Resources:** Magdalena Machnikowska-Sokołowska, Marcin Ciekalski.

**Supervision:** Magdalena Machnikowska-Sokołowska, Katarzyna Gruszczyńska.

**Validation:** Magdalena Machnikowska-Sokołowska, Katarzyna Gruszczyńska.

**Visualization:** Marcin Ciekalski.

**Writing – original draft:** Magdalena Machnikowska-Sokołowska, Marcin Ciekalski, Iga Szymańska, Jakub Mordarski.

**Writing – review & editing:** Magdalena Machnikowska-Sokołowska, Marcin Ciekalski, Iga Szymańska, Jakub Mordarski.

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
