## [Decision Letter · Decision Letter 0]

9 Sep 2024

PONE-D-24-20229Retrospective and statistical analysis of hand and forearm injuries in the Silesian pediatric population – study of post-traumatic X-rays in 2022PLOS ONE

Dear Dr. Ciekalski,

Thank you for submitting your manuscript to PLOS ONE. After careful consideration, we feel that it has merit but does not fully meet PLOS ONE’s publication criteria as it currently stands. Therefore, we invite you to submit a revised version of the manuscript that addresses the points raised during the review process.

**ACADEMIC EDITOR: Please insert comments here and delete this placeholder text when finished.** Be sure to:

We look forward to receiving your revised manuscript.

Kind regards,

Jianguo Wang, PhD

Academic Editor

PLOS ONE

Journal Requirements:

Reviewers' comments:

Reviewer's Responses to Questions

**Comments to the Author**

1. Is the manuscript technically sound, and do the data support the conclusions?

Reviewer #1: Yes

Reviewer #2: Yes

2. Has the statistical analysis been performed appropriately and rigorously? 

Reviewer #1: Yes

Reviewer #2: Yes

3. Have the authors made all data underlying the findings in their manuscript fully available?

Reviewer #1: No

Reviewer #2: Yes

4. Is the manuscript presented in an intelligible fashion and written in standard English?

Reviewer #1: Yes

Reviewer #2: Yes

5. Review Comments to the Author

Reviewer #1: The aim of this work is to conduct a comprehensive retrospective and statistical analysis of hand and distal forearm injuries in Silesian children. I think there are a few points that need to be revised.

1. When analyzing the experimental data, could the authors add a comparison of the ethnicity of all the children who were treated in Silesia? If the experimental data obtained are all from the same ethnicity, the authors should mention it in the conclusion.

2. Since the conclusions are based on statistical analysis, the contribution of this study is insufficient. To address this issue, could the authors provide more specific and practical conclusions, such as recommendations for prevention and treatment?

3. The first sentence in Results Section “The study group consisted of 327 girls and 511 boys who were 11.4 ± 3.6 and 12.0 ± 3.8 years old, respectively (min. 0, max. 18 years).” is ambiguous. How can the minimum age in 12.0 ± 3.8 years old be 0?

4. In Table 1, it is better to add percentage statistics to the numbers of different fractures, for example, the number of fractures in forearm is 98 (xx%).

5. The author should provide detailed descriptions of the tables and figures in the paper to explain what they represent. In addition, authors should place these tables and figures in appropriate locations to make them easier for readers to read.

Reviewer #2: This is a meaningful study, which is helpful for the diagnosis and surgical treatment of fractures. However, I have some questions.

Patients were categorized into five age groups: 0-4 years, 5-8 years, 9-12 years, 12-16 years, and 17-18 years.

What is the theoretical basis for the authors to divide the age into these groups? Are there any references?

The population included by the authors is children, and the authors should add data on the epidemiological characteristics of epiphyseal fractures.

The data included by the authors include hand injuries or fractures. The authors should clarify the difference between the two in the method.

Does the data included by the authors include elbow dislocations in children?

In the data included by the authors, a patient may have multiple fracture sites. The number of fracture sites will be more than the number of fracture sites. Therefore, the authors should clarify the epidemiological characteristics of patients with 2, 3 or even more fractures.

6. PLOS authors have the option to publish the peer review history of their article (what does this mean? ). If published, this will include your full peer review and any attached files.

**Do you want your identity to be public for this peer review?** For information about this choice, including consent withdrawal, please see our Privacy Policy .

Reviewer #1: No

Reviewer #2: No

---

## [Author Response · Author response to Decision Letter 0]

28 Oct 2024

Response to reviewers

Thank you for your time and dedication to review this study.

Reviewer #1: The aim of this work is to conduct a comprehensive retrospective and statistical analysis of hand and distal forearm injuries in Silesian children. I think there are a few points that need to be revised.

1. When analysing the experimental data, could the authors add a comparison of the ethnicity of all the children who were treated in Silesia? If the experimental data obtained are all from the same ethnicity, the authors should mention it in the conclusion.

In the revised version, we included data containing information about ethnicity. However, we did not conduct comparisons or statistical tests due to the small number of individuals with non-Polish ethnicity and to keep our work more clear and concise.

2. Since the conclusions are based on statistical analysis, the contribution of this study is insufficient. To address this issue, could the authors provide more specific and practical conclusions, such as recommendations for prevention and treatment?

In revised version we included new conclusions subchapter.

3. The first sentence in Results Section “The study group consisted of 327 girls and 511 boys who were 11.4 ± 3.6 and 12.0 ± 3.8 years old, respectively (min. 0, max. 18 years).” is ambiguous. How can the minimum age in 12.0 ± 3.8 years old be 0?

In the revised version, we improved the clarity of the sentence, and it now includes information about the mean age.

4. In Table 1, it is better to add percentage statistics to the numbers of different fractures, for example, the number of fractures in forearm is 98 (xx%).

5. The author should provide detailed descriptions of the tables and figures in the paper to explain what they represent. In addition, authors should place these tables and figures in appropriate locations to make them easier for readers to read.

We reviewed our article layout and corrected mistakes according to commentary.

Reviewer #2: This is a meaningful study, which is helpful for the diagnosis and surgical treatment of fractures. However, I have some questions.

1. Patients were categorized into five age groups: 0-4 years, 5-8 years, 9-12 years, 12-16 years, and 17-18 years. What is the theoretical basis for the authors to divide the age into these groups? Are there any references?

In the revised version, we included a reference for the age group division.

2. The population included by the authors is children, and the authors should add data on the epidemiological characteristics of epiphyseal fractures.

The data included by the authors include hand injuries or fractures. The authors should clarify the difference between the two in the method.

In revised version we included clear definition for injury and fracture.

3. Does the data included by the authors include elbow dislocations in children?

In revised version we included a sentence in methodology regarding this information (Our study data doesn’t include elbow dislocations).

4. In the data included by the authors, a patient may have multiple fracture sites. The number of fracture sites will be more than the number of fracture sites. Therefore, the authors should clarify the epidemiological characteristics of patients with 2, 3 or even more fractures.

In the revised version, we included a sentence presenting data on the epidemiology of the multiple fracture group in our study.

---

## [Decision Letter · Decision Letter 1]

26 Nov 2024

PONE-D-24-20229R1Retrospective and statistical analysis of hand and forearm injuries in the Silesian pediatric population – study of post-traumatic X-rays in 2022PLOS ONE

Dear Dr. Ciekalski,

Thank you for submitting your manuscript to PLOS ONE. After careful consideration, we feel that it has merit but does not fully meet PLOS ONE’s publication criteria as it currently stands. Therefore, we invite you to submit a revised version of the manuscript that addresses the points raised during the review process.

**ACADEMIC EDITOR:**

Please carefully address the  comments from two reviewers.The data problem should be clearly addressed.

We look forward to receiving your revised manuscript.

Kind regards,

Jianguo Wang, PhD

Academic Editor

PLOS ONE

Reviewers' comments:

Reviewer's Responses to Questions

**Comments to the Author**

1. If the authors have adequately addressed your comments raised in a previous round of review and you feel that this manuscript is now acceptable for publication, you may indicate that here to bypass the “Comments to the Author” section, enter your conflict of interest statement in the “Confidential to Editor” section, and submit your "Accept" recommendation.

Reviewer #1: (No Response)

Reviewer #2: All comments have been addressed

2. Is the manuscript technically sound, and do the data support the conclusions?

Reviewer #1: Yes

Reviewer #2: Yes

3. Has the statistical analysis been performed appropriately and rigorously? 

Reviewer #1: Yes

Reviewer #2: Yes

4. Have the authors made all data underlying the findings in their manuscript fully available?

Reviewer #1: No

Reviewer #2: Yes

5. Is the manuscript presented in an intelligible fashion and written in standard English?

Reviewer #1: Yes

Reviewer #2: Yes

6. Review Comments to the Author

Reviewer #1: Authors have addressed most of my concerns; however, I have one additional suggestion. Better to briefly list the contributions of this work at the end of the introduction section and to summarize them again in the conclusion section.

Reviewer #2: Thank you for your reply. But I have some other questions.

1. You mentioned that your research data does not include elbow dislocation. However, joint dislocation often occurs in children.

2. Upper limb fractures in children are often accompanied by fractures of the lower limbs or even the spine. Do you collect such data?

3. Did the author count the number of open and closed fractures in children?

4. Did the author count the complications of upper limb fractures in children? For example, are there any children with supracondylar humeral fractures combined with compartment syndrome?

7. PLOS authors have the option to publish the peer review history of their article (what does this mean? ). If published, this will include your full peer review and any attached files.

**Do you want your identity to be public for this peer review?** For information about this choice, including consent withdrawal, please see our Privacy Policy .

Reviewer #1: No

Reviewer #2: No

---

## [Author Response · Author response to Decision Letter 1]

8 Jan 2025

Response to reviewers

Reviewer #1: Authors have addressed most of my concerns; however, I have one additional suggestion. Better to briefly list the contributions of this work at the end of the introduction section and to summarize them again in the conclusion section.

Response to Reviewer Comments

We included shortened list of this work contributions at the end of introduction section as suggested.

Reviewer #2: Thank you for your reply. But I have some other questions.

Response to Reviewer Comments

1. You mentioned that your research data does not include elbow dislocation. However, joint dislocation often occurs in children.

Thank you for raising this point. Unfortunately, the scope of the current study does not include data on elbow dislocation in children. The research was designed with a focus on distal forearm wrist, and hand fractures, and joint dislocations were not within the predefined inclusion criteria.

2. Upper limb fractures in children are often accompanied by fractures of the lower limbs or even the spine. Do you collect such data?

We appreciate the suggestion to consider associated fractures in other anatomical regions. However, our dataset was limited to fractures of the upper limb. No systematic data on co-occurring fractures of the lower limbs or the spine were collected.

3. Did the author count the number of open and closed fractures in children?

The study does not include a detailed classification of fractures as open or closed. While such categorization could enrich the dataset and provide deeper insights into the severity and management implications, it was not part of the current research methodology.

4. Did the author count the complications of upper limb fractures in children? For example, are there any children with supracondylar humeral fractures combined with compartment syndrome?

Complications such as compartment syndrome in children with supracondylar humeral fractures were not analysed in the present study. While we recognize the clinical importance of evaluating such complications, they were outside the scope of this research. Future studies could incorporate this analysis to enhance the understanding of outcomes and complications associated with paediatric upper limb fractures.

---

## [Decision Letter · Decision Letter 2]

23 Jan 2025

Retrospective and statistical analysis of hand and forearm injuries in the Silesian pediatric population – study of post-traumatic X-rays in 2022

PONE-D-24-20229R2

Dear Dr. Ciekalski,

We’re pleased to inform you that your manuscript has been judged scientifically suitable for publication and will be formally accepted for publication once it meets all outstanding technical requirements.

Kind regards,

Jianguo Wang, PhD

Academic Editor

PLOS ONE

Additional Editor Comments (optional):

Reviewers' comments:

Reviewer's Responses to Questions

**Comments to the Author**

1. If the authors have adequately addressed your comments raised in a previous round of review and you feel that this manuscript is now acceptable for publication, you may indicate that here to bypass the “Comments to the Author” section, enter your conflict of interest statement in the “Confidential to Editor” section, and submit your "Accept" recommendation.

Reviewer #1: All comments have been addressed

2. Is the manuscript technically sound, and do the data support the conclusions?

Reviewer #1: Yes

3. Has the statistical analysis been performed appropriately and rigorously? 

Reviewer #1: Yes

4. Have the authors made all data underlying the findings in their manuscript fully available?

Reviewer #1: No

5. Is the manuscript presented in an intelligible fashion and written in standard English?

Reviewer #1: Yes

6. Review Comments to the Author

Reviewer #1: After making revisions, the authors have addressed my concerns and I think this manuscript is ready to be accepted.

7. PLOS authors have the option to publish the peer review history of their article (what does this mean? ). If published, this will include your full peer review and any attached files.

**Do you want your identity to be public for this peer review?** For information about this choice, including consent withdrawal, please see our Privacy Policy .

Reviewer #1: No

---

## [Editor Report · Acceptance letter]

PONE-D-24-20229R2

PLOS ONE

Dear Dr. Ciekalski,

I'm pleased to inform you that your manuscript has been deemed suitable for publication in PLOS ONE. Congratulations! Your manuscript is now being handed over to our production team.

Kind regards,

on behalf of

Dr. Jianguo Wang

Academic Editor

PLOS ONE